# Validation of a Speech Database for Assessing College Students’ Physical Competence under the Concept of Physical Literacy

**DOI:** 10.3390/ijerph19127046

**Published:** 2022-06-08

**Authors:** Rui-Si Ma, Si-Ioi Ng, Tan Lee, Yi-Jian Yang, Raymond Kim-Wai Sum

**Affiliations:** 1Department of Sports Science and Physical Education, Faculty of Education, The Chinese University of Hong Kong, Hong Kong, China; penny@link.cuhk.edu.hk (R.-S.M.); yyang@cuhk.edu.hk (Y.-J.Y.); 2School of Physical Education, Jinan University, Guangzhou 510632, China; 3Department of Electronic Engineering, Faculty of Engineering, The Chinese University of Hong Kong, Hong Kong, China; siioing@link.cuhk.edu.hk (S.-I.N.); tanlee@ee.cuhk.edu.hk (T.L.)

**Keywords:** machine learning, speech database, physical literacy, physical competence

## Abstract

This study developed a speech database for assessing one of the elements of physical literacy—physical competence. Thirty-one healthy and native Cantonese speakers were instructed to read a material aloud after various exercises. The speech database contained four types of speech, which were collected at rest and after three exercises of the Canadian Assessment of Physical Literacy 2nd Edition. To show the possibility of detecting each exercise state, a support vector machine (SVM) was trained on the acoustic features. Two speech feature sets, the extended Geneva Minimalistic Acoustic Parameter Set (eGeMAPS) and Computational Paralinguistics Challenge (ComParE), were utilized to perform speech signal processing. The results showed that the two stage four-class SVM were better than the stage one. The performances of both feature sets could achieve 70% accuracy (unweighted average recall (UAR)) in the three-class model after five-fold cross-validation. The UAR result of the resting and vigorous state on the two-class model running with the ComParE feature set was 97%, and the UAR of the resting and moderate state was 74%. This study introduced the process of constructing a speech database and a method that can achieve the short-time automatic classification of physical states. Future work on this corpus, including the prediction of the physical competence of young people, comparison of speech features with other age groups and further spectral analysis, are suggested.

## 1. Introduction

### 1.1. Physical Literacy

Physical literacy (PL) is a concept that values physical activity (PA) for the individual’s health and active living style throughout the life course [1,2]. The concept has gained greater attention and widespread research interest in the academic community in recent years [3]. It has been accepted in many countries as a valuable concept that can help individuals living healthy lives and, also, as the foundation for lifelong active living [4,5]. Various related sectors and organizations, such as physical education, recreation and public health, are constantly exploring how to develop strategies and policies based on PL [6,7]. The dimensions of PL are assessed in many ways. The most widely accepted definition is the concept proposed by Whitehead, which was built on the dimensions of motivation, confidence, physical competence and knowledge [2]. Researchers continue to emphasize that evaluating PL requires a full range of objective measurements as the way to understand individuals’ PL status and to assess the effectiveness of exercise programs [8].

Currently, there are three PL assessment instruments that combine objective and subjective parameters, including passport for life (PFL) [9], physical literacy assessment for youth (PLAY) tools [10] and the Canadian assessment of physical literacy 2 (CAPL2) [5]. A systematic review on assessments related to PL analyzed 15 measurements and concluded that CAPL2 and PFL had a high standard methodological quality and assessed a wide range of PL elements [11]. CAPL2 also had a high score in cross-cultural validity. CAPL was the first measurement constructed to assess PL through large-scale research and develop a program [12]. It underwent a second edition revision in 2017. More than 100 researchers have validated CAPL with over 11,000 participants’ data [13]. The four domains of physical competence: daily behavior, motivation and confidence and knowledge and understanding, are used to assess the PL level of each participant for targeted interventions. The scores for each dimension are then interpreted separately according to age and sex. The scores are divided into four categories: beginning, progressing, achieving and excelling. Unlike other tools that are used locally, the CAPL has been translated in more than five languages and used internationally [14]. Now, the number of studies using CAPL and CAPL 2nd Edition (CAPL2) continues to grow and has made a significant impact around the world. Therefore, our study chose CAPL2 to assess the physical competence. In CAPL2, physical competence is assessed in three movements, including Canadian agility and movement skill assessment (CAMSA), plank and pacer [13]. The procedure requires multiple trained examiners and often takes more than an hour to complete. Each time, it also requires a labor- and material-intensive test. Due to this, an alternative and simple assessment tool is needed. This study thus attempts to utilize speech signal processing technology to fill this research gap.

### 1.2. Speech While Exercising

The talk test is one way to measure one’s exercise intensity. If one can still speak comfortably, they are within the recommended guidelines for intensity [15]. Therefore, the ability to speak during exercise is affected by the sport’s intensity. This has led researchers to examine how such measurements can be used to objectively assess the intensity of an exercise [16]. Several studies have also explored classifying low and high physical stress in speech [17,18]. Inspired by this, the present study aimed to build an acoustic database, providing data to further support the analysis of acoustic signals under different exercises.

Statistical models are built on parameters, also called features, that are individual measurable properties or characteristics of a phenomenon being observed [19]. In speech recognition, the features for recognizing phonemes include noise rations, lengths of sounds, relative power, filter matches and many others. Previous studies using stair stepper and bicycle stress tests found that speech parameters such as F0 and the percentage of vocal frames increased and decreased, respectively, under the influence of moderate and high exercise intensities [16,20,21,22]. One study also found that the number of inappropriate pauses increased while performing physical exercise at 50% and 75% of the maximal oxygen consumption (VO2 max) [23]. Lately, these speech features have been increasingly investigated for the automatic classification of high- and low-exercise intensity segmentation [24,25]. In 2014, Interspeech, the world’s largest and most comprehensive conference on the science and technology of spoken language processing, organized a session on the topic of speech processing under physical stress [17]. In the challenge, participants were asked to analyze speech signals during various physical loads. More and more studies thereafter have begun to focus on the changes in speech features during exercise. Vowel and breath lengths were used as speech features to classify the exercise intensity for higher and lower levels [25]. Additionally, Mel-frequency cepstral coefficients (MFCCs) were used as features to perform the classification [16]. One study went further and analyzed speech data while running [18]. These studies contributed to the process of analyzing physical states based on speech signal processing. Inspired by previous research, the present study planned to utilize speech data to classify the physical competence measurements of CAPL2. Compared to previous studies that classified two groups, our study was innovative and technically challenging. Specifically, we aimed to utilize the latest classifier models to classify four types of exercises and to develop a novel method for predicting a physical competence score through speech analysis.

## 2. Methods

### 2.1. Participants

Our study sample consisted of healthy, young adults. Participants were recruited online through the Chinese University of Hong Kong (CUHK) system, and all participants were university students. In total, there were 31 (24 males and 7 females) participants who completed the entire study, and no one dropped out. Each of them met the following inclusion criteria: (1) college students and (2) no medical history of cardiovascular disease or pulmonary disease. Written consent was obtained from all participants, and ethical approval was obtained from the Survey and Behavioral Research Ethics Committee of CUHK (SBRE-20-219).

### 2.2. Study Design

This was a cross-sectional study. All exercises were conducted in the outdoor sports field of CUHK. Information on demographics and physical activity was collected. All participants were required to wear heart rate devices throughout the test. Participants were asked to record their resting heart rate and then begin the CAPL2 physical competence test. At the end of each exercise, student helpers recorded the participants’ current heart rate and recorded the speech data. The total length of the speech was about one and a half hours. All helpers were well-trained beforehand to conduct the tests and operate the equipment.

### 2.3. Measurements

**Speech data** was recorded with professional recording equipment (TASCOM DR-44WL) that allowed amplification of the input signal and the simultaneous recording of separate audio channels. The recorder was located 20–50 cm in front of the participant’s mouth. Environmental noise from outdoor venues, such as distant walking and talking sounds, were inevitably included. The gain of the recorder was maximized to keep the noise below −30 dB (relative to the maximum input level), and the sampling rate was set at 44.1 kHz. **The heart rate** was measured continuously by the Polar OH1 sensor worn by the participants while doing the exercises. The score of the physical competence test of **CAPL2** was also recorded by trained helpers according to the CAPL2 guidelines [13]. **IPAQ** was used to collect participants’ general physical conditions [26].

### 2.4. Reading Materials

Participants were asked to read a short text aloud in Cantonese. As shown in Figure 1, the text contained four parts: the first part was the well-known speech study text *The North Wind And The Sun* (Cantonese version) (see Figure 1). The second part was a text containing nouns that were relatively difficult to pronounce. The third part was three long vowel characters, each of which was asked to be pronounced in about three seconds and repeated once. The fourth part was a string of numbers containing different tones of Cantonese, which were also required to be read aloud twice.

## 3. Data Analysis

### 3.1. Speech Features

openSMILE 3.0, an open-source toolkit for speech signals processing, was utilized for speech features extraction [27]. The openSMILE toolkit performs the extraction of acoustic parameters that describe the paralinguistic characteristics of the speech signal. Based on the previous success that these acoustic parameters were able to assess the personality [28], detect a speech-related disease [29] and identify the gender and age [30] of a person, we deployed the acoustic parameter sets defined in eGeMAPS and ComParE to facilitate the classification of the physical load based on speech signals [31,32].

eGeMAPS was built based on the Geneva Minimalistic Acoustic Parameter Set (GeMAPS), which included 62 parameters. On top of that, eGeMAPS contained an equivalent sound level, which led to 26 extra parameters [32]. This total of 88 parameters has often been used in speech processing studies. eGeMAPS has been widely used in speech research as a comprehensive set of features that reflect the emotional properties in speech according to their theoretical meaning and potency. This feature set was intended to provide a common basis for emotion-related speech features and has since become a de facto standard. This study thus used this feature set to participate in this experiment. The low-level descriptors (LLD) in eGeMAPS include frequency-related features such as F0 and formant frequencies, energy-related features such as shimmer and loudness and temporal features such as the rate of loudness peaks. To facilitate a comparison, all LLD and functions were not changed and remained the same as the original version [32].

ComParE included 6373 static features computed from various functions of LLDs [31]. The feature set contains energy-related LLDs, spectral LLDs, sound-related LLDs, functionals applied to LLDs/ΔLLD and functions only applied to LLDs. Such a large number of parameters stabilized this feature set during various experiments. The feature set has been successfully applied to different scenarios to analyze the cognitive load, physical load, emotion, speech-related disease, etc. Similar to eGeMAPS, the ComParE feature set has also often been used in the field of acoustical research, so it is worth using during this task. This study did not make any modifications to the feature set. These two acoustic feature sets were only used for exploring the feasibility of the classification of the physical competence status. In order to compare the use of high-dimensional (ComParE) and low-dimensional (eGeMAPS) parameters in our task, no preprocessing was applied during the experiment.

### 3.2. Statistical Analysis

To increase the sample size and to cope with the feasibility of short-time processing, we split the speech data into segments. The audio was intercepted as a segment from zero seconds until ten seconds later. After that, the next ten seconds were intercepted, with no overlap in between, and so on. The last part with less than ten seconds was discarded from the experiment. In this study, all speech data were randomly divided into a training set, development set and test set in a ratio of 6:2:2. The training dataset was used to fit the model, the development dataset was used to provide an unbiased evaluation of the model fit of the training dataset while tuning the model hyperparameters. The final model fit was provided by the test dataset.

This study used a SVM model to show the possibility of the classification of each physical state. For the SVM model, the training dataset was $\left\{x_{i},\;y_{i}\right\},\;x_{i}\in \;R^{d},\;i=1,\;\ldots ,\;n$ with two labels: $y_{i}\in \left\{-1,+1\right\}$. The SVM would find the optimal hyperplane [33].
$$f\left(x\right)=w^{T}\varphi \left(x\right)+b$$

The training data were separated by solving the following optimization problems:$$min_{w,b}\left(\frac{1}{2}{\left|\left|w\right|\right|}^{2}+C\sum_{i=1}^{n}\;\xi_{i}\right)$$
subject to
$$y_{i}\left[w^{T}\varphi \left(x_{i}\right)+b\right]\geq 1-\xi_{i}\;\;\;\;\;\;\;\;\;\;\;\;\;\;\;i=1,\;\ldots ,\;n\;\xi_{i}\geq \;0,\;i=1,\;\ldots ,\;n$$
where w is the normal vector of the hyperplane, C and b are real numbers and $\varphi \left(.\right)$ is a kernel function. When the error occurs, the corresponding $\xi_{i}$ must exceed the unity. The upper bound on the number of training errors is $\Sigma_{i}\xi_{i}$. Extra cost $C\sum_{i=1}^{n}\;\xi_{i}$ for the errors is added with a chosen user C. $f\left(x\right)=sign\left(w^{T}\varphi \left(x\right)+b\right)$ will classify unknown data as positive and negative.

The experiment was attempted twice, The first one was a 4-class support vector machine (SVM) with a linear kernel [34]. A grid search was utilized to find the optimized gamma *g* and cost *c* parameters of the SVM algorithm. The second one included two stages, which first trained a 3-class SVM model to classify rest, moderate and vigorous exercises, and then trained a 2-class SVM model to classify CAMSA and Plank. In order to perform the best assessment, the classification experiments were further carried out with an arrangement of 5-fold cross-validation for examining the 3-class SVM [35]. Unweighted average recall (UAR) was used to assess the accuracy of the classification, and Cohen’s Kappa coefficient (*k*) was utilized to measure the reliability of the results [36]. All analyses were performed using scikit-learn 1.0.1 [37].

## 4. Results

All participants provided valid data for the CAPL2 score and speech while exercising. No one withdrew in the middle of the experiments. The results of the descriptive analysis on the participants’ characteristics are shown in Table 1. The participants were in relatively good physical condition.

The unweighted average recalls (UAR) of the best-performing classifiers obtained according to the SVM are shown in Table 2 and Table 3. The confusion matrix was used to describe the performance of the classification model. Each number represents the results of the sound segment predictions, and the numbers in bold represent the correct predictions. We compared the classification effects of two different feature sets. In terms of feature set selection, eGeMAPS performed lower than ComParE overall. The results of the two-stage modeling (UAR_ComParE_ = 0.65, *k* = 0.30) was better than the one-stage (UAR_ComParE_ = 0.54, *k* = 0.33). The major issue of classification was that it was difficult to distinguish between two exercises (CAMSA and Plank) that were of the same moderate intensity. Even under the binary classification model, two exercises of the same intensity still have a great similarity in their speech features. To further test the accuracy of the three-class model, five-fold cross-validation was performed (Table 4). With more training and testing data, the results for both feature sets performed well (UAR_ComParE_ = 0.70, *k* = 0.30 and UAR_ComParE_ = 0.70, *k* = 0.34). All the models performed well in the binary classification. The UAR result for the resting and vigorous state of the two-class model running with the ComParE feature set was 0.97, and the UAR for the resting and moderate state was 0.74.

## 5. Discussion

How to evaluate individuals’ physical competence has been an important topic in many sport-related studies. Our study showed a new way of exploring CAPL2 in the PL assessment tool, as an example. Since human breathing is related to lung function, and speech shares a system with breathing, the voice of a person after exercise is closely related to his or her own exercise intensity, body functions such as lung function and respiratory system. In automatic speech recognition (ASR), speech data under physical stress is not applicable to statistical models trained from speech at rest [38]. A database of speech while exercising is essential in the in-depth study of the changes in the speech characteristics of individuals under physical stress. Our study provided a database of speech while exercising, which included recordings after three different exercises, as well as in the resting state. Our study addressed the question of “what exercise the speaker is performing” and, thus, set the foundation for more predictive research in the future.

### 5.1. Classification of Different Physical States

There is a certain pattern of changes in individual’s speech characteristics under different physical stresses [20,21,22]. Based on the changes in the speech parameters, physical activity can be broadly predicted as high or low intensity [17]. However, no research has been conducted to investigate how to distinguish more than two physical activities with similar exercise intensities. In this study, we intended to deal with the problem. We utilized the speech data collected by CAPL2, a PL assessment tool that includes three exercises, to build two statistical models for predicting different physical states, which were (1) the resting state, (2) CAMSA, (3) plank and (4) pacer.

From the results, the overall classification using the ComParE feature set was better than eGeMAPS. This may be because our chosen exercises did not only include running, climbing ladders and other relatively fixed movements, such as in previous studies [20,21,22]. CAMSA contains agility and balance movements, and pacer contains folding actions. The effects of these complex movements on speech are not as direct and significant as that of fixed movements. Therefore, the feature set with the largest set of features was superior to the small feature set during the classification performances. However, the three-class SVM results of the cross-validation showed that a similar accuracy could be obtained for both feature sets, which indicated that a larger feature set could be fitted to compensate for a small sample size. A small feature set can also perform well when the sample size is large enough.

In terms of the stages of classification, previous studies could only dichotomize the physical status or other related indicators [16,17,18]. Based on this, we first explored a four-class prediction. With the four-class model, different exercises at the same intensity did not show differences in their speech features. This made the task more difficult. We then tried a two-stages approach, first using a three-class SVM model to distinguish between moderate intensity, higher intensity, and the resting state. Then, a two-class classification model was used to distinguish exercises at a moderate intensity. The results of the two-stages approach were better than the first one.

Notably, almost all the speech at rest could be distinguished from those of vigorous intensity, and the distinction between vigorous and moderate intensity was also well-performed. The accuracy of the distinction between rest and moderate intensity exercise was reduced, partly because the changes in the speech features were not obvious under moderate intensity exercise. Another main source of false predictions comes from the fact that the data were analyzed in ten-second fragments. Changes in speech features from the exercise intensity may be difficult to reflect on over a relatively short period of time. Under exercise stress, people need to inhale more oxygen to maintain body consumption, and because speech and breathing share a common system, breathing will take up part of the time of speech [23]. Some participants chose to increase the number of pauses, while others chose to take one prolonged pause after a long sentence. This resulted in a number of segments that did not include motor stress features in the data. In addition, some of the participants also produced pauses, a higher pitch, and faster speech speeds at the beginning of their reading due to motor stress, but during the reading process, their physical fatigue was gradually recovered from, so their voices gradually returned to their usual states, which led to some of the data being misclassified as moderate intensity or even calm speech. Future studies should select 15 or 20-s segments and reduce the length of the reading material to preserve the richer speech features under exercise stress.

These findings will contribute to future ASR research, and distinguishing between different exercise types at the same exercise intensity will become a more interesting challenge. In addition, speech data are sensitive and susceptible to external disturbances, such as noise or human voices in the environment, so previous studies chose indoor exercises in order to collect data [16,18,24]. The speech data in this study were all from outdoor exercises, and the experimental results were still satisfactory in the presence of ambient noise. This also showed that the prediction model has more potential in resisting noise.

### 5.2. Applications of the Speech Database

Using the physical competence test in CAPL2, we collected speech data containing four markers. In addition, we recorded both the participants’ scores for different exercises and their heart rates after exercise. Previous research stated that physical stress brings about changes in speech [20]. Physical stress often refers to the intensity of the exercise. As the intensity of the exercise gradually increases, people need more oxygen [39]. Since breathing shares a system with speech, people tend to pause more during speech and use an increased range of intonation after exercising [23]. Therefore, using speech features to distinguish the exercise intensity has an innate advantage [16,17,18]. The study conducted at the same time analyzed the variations of the fundamental frequency in the motion state [40]. In this study, a two-stage statistical model was developed using the SVM algorithm to tackle this issue. Based on the results of this study that predicted similar exercise intensities, it is reasonable to assume that speech analysis can incorporate more relevant metrics into the scope of the experiments, such as predicting physical competence.

Although this database was primarily designed to produce future predictions of the physical competence scores in CAPL2, the speech data includes four types of physical states, which can also be useful for more relevant research. First, the corpus contains four parts, and this study considers them as a whole for the statistics. In future studies, the variations of each part of the contents can be analyzed in more depth, such as whether the features of the vowel part or the number part are better than the other parts. Second, in order to minimize the errors, all participants involved in the experiment were young and healthy, and they did not have any impairment to reading aloud. Based on the existing experimental procedure, it is possible to provide a valid comparison for people of different ages and people with expression disorders in the future. Third, the two feature sets used in this study are fixed and have not been modified in any way. Further research can adjust the feature sets or combine other feature sets to explore a better fit. Fourth, we only cut the sample data into 10 s and compared the results in the segment levels, but future studies could consider using the utterance level and test other segment levels. This speech database provides a good basis for this related research and applications. The automatic detection of the physical state and related parameters is feasible using an approach such as machine learning and deep learning. Moreover, automated measurement tools can speed up physical, and even physiological, assessments, thus saving time and improving the accuracy. In the long run, related research can provide smart solutions for the sports, as well as health, industries.

## 6. Limitations

A limitation of this study is that it was based on the speech data of Cantonese pronunciation. Due to the differences in pronunciation between languages, even when translated into another language with the same meaning, it is difficult to gain a good comparability because of these differences. Therefore, similar studies in other languages cannot directly use the statistical model of the current study, and the results of this study may not be generalized. Second, this study did not recruit enough female participants, and future research should explore whether a classification model predicting female voices would be significantly different from that of males. Third, the low performance on Plank may be caused by the similarity of the speech features under the same exercise intensity. Further study could use a larger amount of training data to test the results.

## 7. Conclusions

This paper introduced a speech corpus of three physical competence tests based on CAPL2. We then utilized the database to conduct a two-stage classification experiment. The results were in line with previous studies [16,24] and showed that the automated classification of exercises can be achieved through acoustic features. This study also opened up avenues for further investigations—for instance, the identification of not only the intensity of exercise but, also, the different types of exercise. Our study set the foundation for further research of speech under physical stress. Meanwhile, its assistance in the assessment of physical competence is equally important. Future works on this corpus, including the prediction of the physical competence of young people, comparisons of speech features with other age groups and further spectral analyses, are suggested.

## Figures and Tables

**Figure 1 ijerph-19-07046-f001:**
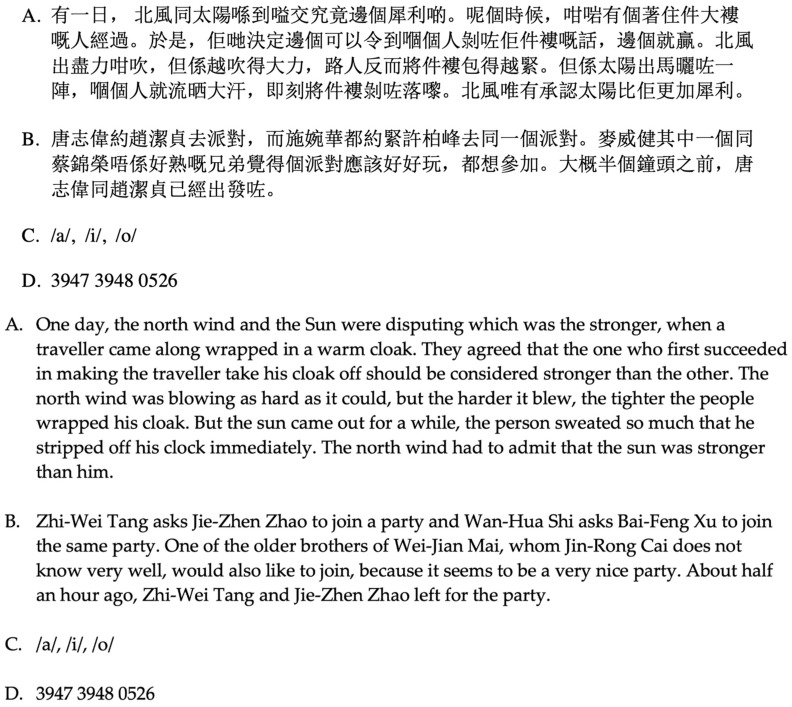
Reading materials.

**Table 1 ijerph-19-07046-t001:** Participants’ characteristics (n = 31).

	Mean ± SD
Age (year)	18.97 ± 0.91
Weight (kg)	66.54 ± 10.67
Height (cm)	173.55 ± 7.25
BMI (kg/m^2^)	22.00 ± 2.52
Vigorous PA time (min/day)	113.71 ± 49.26
Moderate PA time (min/day)	65.32 ± 52.33
Walking time (min/day)	63.87 ± 43.10
Sedentary time (min/day)	326.13 ± 133.88

**Table 2 ijerph-19-07046-t002:** Results of the 4-class SVM.

	eGeMAPS					ComParE				
	CAMSA	Pacer	Plank	Rest	Recall	CAMSA	Pacer	Plank	Rest	Recall
CAMSA	**21**	0	3	19	0.49	**17**	5	8	13	0.40
Pacer	11	**28**	1	2	0.67	11	**25**	5	1	0.78
Plank	19	1	**8**	12	0.20	7	1	**15**	17	0.47
Rest	8	2	12	**32**	0.59	7	1	4	**42**	0.58
Total					0.49					0.54
				*k* = 0.32					*k* = 0.33	

**Table 3 ijerph-19-07046-t003:** Results of the two stages SVM.

	eGeMAPS (3-Class SVM)				ComParE (3-Class SVM)			
	Moderate	Vigorous	Rest	Recall	Moderate	Vigorous	Rest	Recall
Moderate	**51**	2	30	0.61	**52**	6	25	0.63
Vigorous	12	**29**	1	0.69	17	**25**	0	0.60
Rest	24	2	**28**	0.52	16	1	**37**	0.69
				0.60				0.64
			*k* = 0.25				*k* = 0.33	
	**eGeMAPS (2-class SVM)**				**ComParE (2-class SVM)**			
	**CAMSA**	**Plank**		**Recall**	**CAMSA**	**Plank**		**Recall**
CAMSA	**28**	15		0.65	**27**	16		0.63
Plank	22	**18**		0.45	13	**27**		0.68
				0.55				0.65
Total				0.58				0.65
		*k* = 0.10				*k* = 0.30		

**Table 4 ijerph-19-07046-t004:** Results of the cross-validation.

	eGeMAPS (3-Class SVM)				ComParE (3-Class SVM)			
	Moderate	Vigorous	Rest	Recall	Moderate	Vigorous	Rest	Recall
Moderate	**317**	34	67	0.76	**310**	37	72	0.74
Vigorous	68	**177**	3	0.71	67	**174**	7	0.70
Rest	113	9	**131**	0.52	82	6	**165**	0.65
Total				0.70				0.70
			*k* = 0.30				*k* = 0.34	

## Data Availability

The database is available on the Harvard Dataverse, Ma, Rui Si, 2022, “Speech database for classifying different exercises”, https://doi.org/10.7910/DVN/SWOCEZ (accessed on 6 June 2022).

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
