# Peer review of "Validation of a Speech Database for Assessing College Students’ Physical Competence under the Concept of Physical Literacy"

_ijerph, 2022, doi:10.3390/ijerph19127046_

Round 1

Reviewer 1 Report

The authors present a speech  database for assessing physical competence, discussing motivation and characteristics of the collected data; moreover, an experimental section discusses some classification results. The work is interesting for the novelty of the approach and the setting; some issues are related to the main objective of the paper: it is not clear if the work focuses on the dataset description or on the classification results; on the one hand import details about the speech data are not sufficiently highlighted (e.g.,  amount of the audio material, the measured SNRs, possible forced alignments); on the other, the experimental setting doesn't address more recent approaches based on neural classifiers. Indeed, the Computational Paralinguistics Challenge represents a recurrent initiative addressing several number of paralinguistic phenomena so specific experiments (eg. ablation study) could improve the quality of the classification experiments as well as the comparison with recent SoTA neural approaches. Finally, an earlier paper on the same corpus could be inserted in the references ("Acoustical Analysis of Speech Under Physical Stress in Relation to Physical Activities and Physical Literacy").

Author Response

The authors present a speech  database for assessing physical competence, discussing motivation and characteristics of the collected data; moreover, an experimental section discusses some classification results. The work is interesting for the novelty of the approach and the setting; some issues are related to the main objective of the paper:

Point 1: it is not clear if the work focuses on the dataset description or on the classification results; on the one hand import details about the speech data are not sufficiently highlighted (e.g.,  amount of the audio material, the measured SNRs, possible forced alignments); on the other, the experimental setting doesn't address more recent approaches based on neural classifiers. Indeed, the Computational Paralinguistics Challenge represents a recurrent initiative addressing several number of paralinguistic phenomena so specific experiments (eg. ablation study) could improve the quality of the classification experiments as well as the comparison with recent SoTA neural approaches.

Response 1: Thank you for your suggestion. We intended to create a database of speech under physical stress for analyzing various speech features. This study demonstrates one of the classification possibilities. For amount of the audio material, we’ve added the information in line 105. The information of SNRs was mentioned in line 112-113 (“The gain of the recorder was maximized to keep the noise below -30 dB (relative to the maximum input level)”). For forced alignments, this study didn’t separate each reading part, we used speech data as a whole and split it into segments of 10s. We also believe that if the four reading parts are analyzed separately there may be different results, and therefore in the second paragraph of 5.2 it is mentioned that future research can be done further in this area. (Line 272-275) For more recent approaches based on neural classifiers, as mentioned before, we intended to show a possibility to classify physical states. Limited by the size of the data and the singularity of the speaker's background, some other approaches such as neural networks were not tested here.

Point 2: Finally, an earlier paper on the same corpus could be inserted in the references ("Acoustical Analysis of Speech Under Physical Stress in Relation to Physical Activities and Physical Literacy").

Response 2: Thank you for your suggestion. We’ve added the citation in line 266-267 and line 387.

Reviewer 2 Report

The manuscript approaches the assessment of physical competence by means of speech analysis and classification, providing a speech database for levels of physical stress.

It is rather surprising the unweighted average recall (UART) for the resting and moderate states was only 74%. What are the causes of this decreased sensitivity as compared the resting and vigorous state classification?

In part of your analysis you select "two lower-case feature sets". One wonders whether they are text or speech features.

Author Response

The manuscript approaches the assessment of physical competence by means of speech analysis and classification, providing a speech database for levels of physical stress.

Point 1: It is rather surprising the unweighted average recall (UART) for the resting and moderate states was only 74%. What are the causes of this decreased sensitivity as compared the resting and vigorous state classification?

Response 1: Thank you for your suggeston. We’ve added the explanation in line 240-242. Since there was already a part of the explanation in line 242-252, we’ve merged them together.

Point 2: In part of your analysis you select "two lower-case feature sets". One wonders whether they are text or speech features.

Response 2: Thank you for your suggestion. We’ve added more explanations and reorganized them for a better understanding. (Line 129-132)

Reviewer 3 Report

The manuscript treat about the developing of a speech database for assessing one of the elements of physical literacy - physical competence, and about the implementation of an AI technique for classification purposes. It is a very interesting research subject, and definitely it is inside of the journal scope. IMO, the major contribution is the speech DB creation, but I observed some improvements in the test development, explaining with more details some points. I have the following comments:

The abstract is not clear and it needs significant improvements.

The content of the paper is well organized, but maybe some improvements can be performed, such as the inclusion of an introduction text before to start a new subsection, for example: section 1 and subsection 1.1, but this is only a personal preference.

Include some mathematical support of the techniques used for a better understanding (no equation along the paper)

What were the criteria to choose the openSmile 3.0 tool and specially the feature sets eGeMAPS and ComParE.  A brief description of the main parameters of these selected features should be presented, not describing each one but the main group of parameters (temporal, spectral, number of mfcc and others…).  Thus, In the Result Section it is possible to analyze the relevance of some features.

Another doubt is why a pre-processing data step was not performed, for example a reduction technique of input parameters.  

In the methodology section of statistical analysis, it is stated that the speech samples were dividied into 136 segments 10s, What 10 seconds?

It is ideal that the percentage of men and women in the total number of volunteers/participants be almost equal, but authors stated as limitation of the works that is fine.

This specific section need more details, it is too summarized, important information about the training and test phases are mentioned that were not properly justified. Why 5-fold validation were use (and not k=10)? because of number of samples of each class?

The performance metrics need to be explained or at least give some references for a better understanding.

I understand that because authors use a new DB to perform the tests, no performance comparisons with other solutions can be presented. Do other solutions reach similar performance results ? Authors use only SVM, what other ML techniques are used to develop similar solutions in the current literature?

Author Response

The manuscript treat about the developing of a speech database for assessing one of the elements of physical literacy - physical competence, and about the implementation of an AI technique for classification purposes. It is a very interesting research subject, and definitely it is inside of the journal scope. IMO, the major contribution is the speech DB creation, but I observed some improvements in the test development, explaining with more details some points. I have the following comments:

Point 1: The abstract is not clear and it needs significant improvements.

Response 1: Thank you for your suggestion. We have reorganized the content of the abstract to make it clearer.

Point 2: The content of the paper is well organized, but maybe some improvements can be performed, such as the inclusion of an introduction text before to start a new subsection, for example: section 1 and subsection 1.1, but this is only a personal preference.

Response 2: Thank you for your suggestion. We believe that keeping a uniform style will better convey the content of the article. but thank you again for the great idea.

Point 3: Include some mathematical support of the techniques used for a better understanding (no equation along the paper)

Response 3: Thank you for your suggestion. We’ve added the SVM equation for a better understanding. (Line 164-176)

Point 4: What were the criteria to choose the openSmile 3.0 tool and specially the feature sets eGeMAPS and ComParE.  A brief description of the main parameters of these selected features should be presented, not describing each one but the main group of parameters (temporal, spectral, number of mfcc and others…).  Thus, In the Result Section it is possible to analyze the relevance of some features.

Response 4: Thank you for your suggestion. We’ve added criteria of choosing the opensmile 3.0 tool in line 129-132. We’ve also added three references 28,29, and 30. For two feature sets that this study used, we’ve added the explanations why this study chose these two feature sets. (Line 137-140 and line 149-151) For further analyze the relevance of features, this study only intended to introduce a novel database and show a possible way to classify each label. More research could be done based on this study. (Line 278-279)

Point 5: Another doubt is why a pre-processing data step was not performed, for example a reduction technique of input parameters.  

Response 5: Thank you for your suggestion. We intended to compare the use of high-dimensional (ComParE) and low-dimensional (eGeMAPS) in our task. Thus, no pre-processing is applied in the experiment. We’ve added the explanation for a better understanding. (Line 152-154)

Point 6: In the methodology section of statistical analysis, it is stated that the speech samples were dividied into 136 segments 10s, What 10 seconds?

Response 6: Thank you for your suggestion. We’ve added detailed explanation of the ten second segments. (Line158-160)

Point 7: It is ideal that the percentage of men and women in the total number of volunteers/participants be almost equal, but authors stated as limitation of the works that is fine.

Response 7: Thank you for your suggestion. We plan to include more participants and a balanced male to female ratio in future studies.

Point 8: This specific section need more details, it is too summarized, important information about the training and test phases are mentioned that were not properly justified. Why 5-fold validation were use (and not k=10)? because of number of samples of each class?

Response 8: Thank you for your suggestion. We’ve added more information of training and test phases for a better understanding. (Line 161-163). For 5-fold validation, we chose it empirically. Given we have limited speech data, we must ensure the amount of test data in each fold is reasonable and that the evaluation metric is not too sensitive due to the scarcity of test samples.

Point 9: The performance metrics need to be explained or at least give some references for a better understanding.

Response 9: Thank you for your suggestion. We’ve added the explanation for a better understanding. (Line 191-192)

Point 10: I understand that because authors use a new DB to perform the tests, no performance comparisons with other solutions can be presented. Do other solutions reach similar performance results ? Authors use only SVM, what other ML techniques are used to develop similar solutions in the current literature?

Response 10: Thank you for your suggestion. For more recent approaches based on neural classifiers, we intended to show a possibility to classify physical states in this study. Limited by the size of the data and the singularity of the speaker's background, some other approaches such as neural networks were not tested here.

Round 2

Reviewer 3 Report

Authors anwer the main concerns presented in the first round of revision.